# Is High Folic Acid Intake a Risk Factor for Autism?—A Review

**DOI:** 10.3390/brainsci7110149

**Published:** 2017-11-10

**Authors:** Darrell Wiens, M. Catherine DeSoto

**Affiliations:** 1Department of Biology, University of Northern Iowa, Cedar Falls, IA 50614, USA; 2Department of Psychology, University of Northern Iowa, Cedar Falls, IA 50614, USA; cathy.desoto@uni.edu

**Keywords:** folic acid fortification, unmetabolized folic acid, autism spectrum disorder, neural development

## Abstract

Folate is required for metabolic processes and neural development. Insuring its adequate levels for pregnant women through supplementation of grain-based foods with synthetic folic acid (FA) in order to prevent neural tube defects has been an ongoing public health initiative. However, because women are advised to take multivitamins containing FA before and throughout pregnancy, the supplementation together with natural dietary folates has led to a demographic with high and rising serum levels of unmetabolized FA. This raises concerns about the detrimental effects of high serum synthetic FA, including a rise in risk for autism spectrum disorder (ASD). Some recent studies have reported a protective effect of FA fortification against ASD, but others have concluded there is an increased risk for ASD and other negative neurocognitive development outcomes. These issues are accompanied by further health questions concerning high, unmetabolized FA levels in serum. In this review, we outline the reasons excess FA supplementation is a concern and review the history and effects of supplementation. We then examine the effects of FA on neuronal development from tissue culture experiments, review recent advances in understanding of metabolic functional blocks in causing ASD and treatment for these with alternative forms such as folinic acid, and finally summarize the conflicting epidemiological findings regarding ASD. Based on the evidence evaluated, we conclude that caution regarding over supplementing is warranted.

## 1. Introduction

As a methyl group donor, folate is necessary for the normal processes of metabolism, reproduction and development. In addition, epigenetic regulation of the genome via histone methylation and acetylation depends on folate for stem and progenitor cell differentiation. Folate has important roles in methionine synthesis [1]. Its sufficient presence also assists in avoiding the accumulation of excess levels of serum homocysteine, which are associated with vascular inflammation and damage, and with teratogenic malformations (reviewed in [2]). In sum, folate is crucial to development. It is beyond dispute that supplementation of the diet of women during pregnancy with multivitamins that include folic acid (FA, synthetic supplement pteroylmonoglutamic acid) in order to prevent neural tube defects (NTDs) has been a successful public health program [3,4,5,6,7], although a critical review of the literature has raised some doubts about whether FA itself prevents NTDs [8]. However, FA has been promoted for consumption before and during pregnancy through the use of vitamin tablets or pills, and is also now present in grain-based foods because of fortification, adding to the natural folates in the diet. Thus, levels of FA have increased greatly over the past generation. Recent epidemiological studies have reported some evidence that also links FA supplementation to slight protective effects against autism spectrum disorder (ASD) [9,10] and perhaps other adverse complications associated with pregnancy or neural development of offspring ([11], and see reviews in [12,13]). However, other studies have reported an increased risk for ASD [14,15] and for hindered neurocognitive development in children of mothers who used dietary supplements of synthetic FA [16]. Moreover, some published reviews and reports have raised numerous health issues concerning FA fortification [17,18,19,20]. Here, we outline the reasons excess FA supplementation is a concern, review the history and effects of supplementation, examine the effects of FA on neuronal development using in vitro studies, and then summarize the conflicting epidemiological findings regarding autism.

## 2. Unmetabolized Folic Acid

Folate from natural food sources consists of primarily formyl-tetrahydropteroylglutamates. Folic acid used in supplements is pteroylmonoglutamic acid [21]. They are not identical. The supplemented FA is the monoglutamate form of folate, and unlike the food form, it appears to be initially metabolized in the liver rather than the intestine [22]. Human dihydrofolate reductase (DHFR), the enzyme that converts it to useable dihydrofolate and tetrahydrofolate, has relatively low and variable activity in the liver, especially when compared to other mammals [23]. This sets up the possibility that high intake of FA may result in unmetabolized FA in the blood stream. To be clear, the concern is not an excess of folates in general, but that high levels of pteroylmonoglutamic acid, which depend on liver-based metabolism, will result in high levels of unmetabolized and non-useful FA in the blood, and that this may be harmful (see [22] for an overview). Because there are individual differences in DHFR activity in humans (e.g., C667T variants), and because humans, as a species, have low activity of this enzyme, the competition for binding enzyme is potentially relevant [23], especially so in some persons.

Dietary supplementation coupled with pervasive fortification of grain-derived foods with synthetic FA beginning in the late 1990s may have created a demographic with high serum levels of unmetabolized FA and high erythrocyte FA concentration (summarized in [20,24]). This is not theoretical, but has been demonstrated in a dose-dependent manner when levels of intake are over 200 mg per day [25]. FA is detectable in fasting serum of a majority of subjects tested, and the proportions with detectable levels have risen since fortification programs were initiated [26,27], even becoming detectable in umbilical cord blood [28]. When food supplementation began, models suggested that supplement levels would be approximately 100 mcg per day, but the actual increase has exceeded 215 mcg, double that estimate [29]. Consumption of more than 1 mg of folic acid appears to reliably result in unmetabolized FA, even if the doses are spaced apart. Sweeney and colleagues [25] investigated the effects of consumption using varying dosage schedules. The cumulative amount was what mattered. They administered FA to participants in five equal doses (of 200 mcg) across a day. All participants showed unmetabolized FA circulating after the second, third, fourth and fifth doses, with the highest levels (up to 5 mcg/L) after the final dose [25]. This raises concerns about detrimental effects of high serum synthetic FA [30]. These include effects on the enzyme dihydrofolate reductase [23], regulation of folate uptake in renal and intestinal epithelia [31], reduced cytotoxicity of natural killer T cells in postmenopausal women [27], disregulation of gene expression in lymphoblastoid cells [32], and cytotoxicity to neural tissues and mental health (reviewed in [33,34]). In addition, evidence shows that high FA intake is associated with an increase in incidence of twin births, body fat mass and insulin resistance in offspring, increased risk of colorectal cancer, and other adverse outcomes (reviewed in [20]).

## 3. Neural Development

Neural development is sensitive to maternal metabolic conditions [35,36], and there is ample evidence that excessive dietary FA could be detrimental to neural development. In experiments with rats, Girotto et al., 2013 [37] showed that supplementation of pregnant mothers’ diets with large doses of FA caused changes in brain synaptic transmission and higher susceptibility to seizures in their offspring. Similarly, Barua et al., 2014 [38] demonstrated that mice born to mothers fed a diet high in FA exhibited changes in gene expression in the cerebrum (including many genes involved in development), and also caused behavioral differences that included more ultrasonic vocalization, more anxiety, and hyperactivity. This group also carried out a microarray analysis of cerebellum gene expression in offspring of similarly treated mice [39]. They found that exposure to the higher dose FA diet during gestation caused dysregulation of expression of many genes, including several transcription factors, imprinted genes, neuro-developmental genes and genes associated with ASD in humans. In a still more recent follow-up study Barua et al., 2016 [40] examined such maternally exposed mouse offspring for evidence of epigenome influence by the high FA diet. They found that the cerebellar DNA methylation profile had distinct methylation patterns of CpG and non-CpG sites that were highly sex-specific, and that included several genes that operate in neuronal pathways. These results suggest that moderation of FA supplementation is warranted. As the authors write, “higher amounts of FA may perturb the epigenetic network. These studies highlight the fact that the consequences of such changes on imprinted and candidate autism susceptible genes may be of concern” ([40], p. 284). Such evidence that epigenetic regulation of neural development in offspring is vulnerable to maternal dietary FA is alarming. However, these studies using rodents saw the effects with maternal diets in a range of FA approximately ten-fold higher than is recommended for normal pregnant women (a level chosen because women with a history of NTD affected pregnancy have been prescribed supplementation at ten-fold higher).

The possibility that unmetabolized FA directly causes changes in the behavior of neurons in some neural circuits during development has been investigated by Wiens et al., 2016 [41]. This research utilized a chick embryo dorsal root ganglion (DRG) culture system in which the motility behavior of developing neurons along with their differentiation was investigated. As with any neuron, these DRG cells extend thin, straight cytoplasmic processes (neurites) led by growth cones. The exploratory growth cones drive and instruct the directional elongation of the neurites until they establish contact with target neurons to build neural networks. This is followed by the accumulation of synaptic vesicles that will release neurotransmitters to establish communication. The motility behavior of the growth cone is complex, with dynamic assembly and remodeling of the actin microfilament and microtubule cytoskeleton within (reviewed in [42]). Particularly important are the very fine filapodia (microspikes) that extend from the periphery. These establish adhesion and traction and possess many surface receptors and internal molecular signaling networks that respond to guidance cues, including soluble factors, surfaces of other cells, and extracellular matrix binding sites (reviewed in Lowery and Vactor, 2009 [43] Cammarata et al., 2016 [44]).

Wiens et al. [41] cultured DRGs from chick embryos for 36 h and then used time-lapse image capture and minute-by-minute motility analysis to compare the behavior of growth cones and neurites before and after addition of FA to the medium in the dish on the warmed stage of a microscope. They also fixed and immunostained other sample cultures to reveal the presence of neural networks with synaptic vesicles, and to enumerate and measure neurite lengths. They found that the number of neurites sent out from neurons was not affected, but there was a dose-dependent relationship between neurite length and FA concentration: a significant inhibition of outward extension of neurites. Their median length was shorter. In addition, they reported that the average total of stained synaptogenic areas surrounding each cultured DRG was reduced significantly. Finally, they accumulated data on area changing of the growth cones (dynamic behavior) during the time-lapse image capture, and this data showed a reduction within thirty minutes. Although this study used FA concentrations that were (as with the rodent studies) quite high, and should not be extrapolated to the levels observed in humans, the findings support the idea of a theoretical direct effect. In a published commentary, Wiens [45] has suggested that FA may be involved in a direct competition for binding to the important *N*-methyl-d-aspartate (NMDA) receptor in the synapse. This receptor is known to be important not only in the post-synaptic membrane but also in the presynaptic membrane in growth cones and their filapodia during the development process [46]. The chemical structure of FA contains glutamate at one end, the abundant neurotransmitter that is involved in opening the NMDA receptor channel for ion fluxes. Altering the calcium ion concentration in neurons would have profound effects on neural development because local cytoplasmic calcium ion concentration is a key regulator of neurite extension, synapse formation and synaptic and dendritic spine maturation (see review in Ebert & Greenberg, 2015 [47]). In sum, evidence is strong that increasing FA beyond the recommended amounts is not without risk (Table 1).

## 4. Metabolic Abnormalities and Autoantibodies in Autism

A large number of recent studies of ASD and other psychiatric disorders have revealed associations with metabolic abnormalities related to folate metabolism. These associations have been categorized into five groups: immune disregulation, inflammation, oxidative stress, mitochondrial dysfunction and environmental toxicant exposures (reviewed in [50,51]). Genes in the folate enzymatic pathway and folate receptor genes do exhibit polymorphisms that affect folate metabolism, and the administration of a type of folate that does not require enzymatic reduction by dihydrofolate reductase (DHFR) can improve the symptoms. For example, folinic acid has been used to correct the folate deficiency in cerebral folate deficiency disorder (a condition in which CSF folate level is abnormally lower than blood levels because of disturbed folate transfer across the blood-brain barrier), restoring CSF levels and improving psychomotor symptoms that included ASD (shown in case follow-up) in a group of five children [52]. A more recent study of 48 children with ASD with language impairment showed that folinic acid treatment for 12 weeks improved verbal communication for 23 children compared to 25 receiving placebo, though the authors caution that the study should be considered preliminary [53].

Interestingly, it has been shown that the failure of folate transfer to CSF in this syndrome is rarely because of mutations in the gene for the folate receptor FRα [54] but is most often due to blockage of the receptor by autoantibodies. This led to the recognition that, in many children with ASD, autoantibodies to FRα are present, and are responsible for the disrupted folate metabolism [55,56]. The prevalence in children with ASD of autoantibodies of a type that either blocks binding of folate to FRα or a type that binds to FRα and evokes an immune reaction is high, estimated to be as high as half or more of children tested [56,57]. Furthermore, a pilot study done with rats has demonstrated that exposure to rat FRalpha antibodies during pregnancy and weaning resulted in severe behavioral deficits in offspring [58]. These discoveries have made treatment with folinic acid and other therapeutic routes related to metabolic abnormalities and autoantibodies feasible. They also have implications for the public health issue of widespread supplementation using FA. Clearly, the continuation of high levels of consumption of FA, if it cannot be metabolized, would be ineffectual.

## 5. Autism Risk Studies

As indicated in Table 1, some important empirical findings suggest FA may be relevant to neurodevelopment abnormalities. Moreover, the observation that increases in neurodevelopment disorder diagnoses (i.e., autism) temporally correlate with increased FA supplementation has been made [19]. Thus, some researchers have attempted to see if FA supplementation may increase autism risk. The results have been contradictory. Surén et al., 2013 [10] reported a protective effect. Schmidt et al., 2012 [9] also reported a protective effect. On the other hand, DeSoto and Hitlan, 2012 [14] reported an increased risk, as did Raghavin et al., 2016 [15]. Given the importance of this question, each study will be separately considered in this review.

### 5.1. Surén et al., 2013 [10]

Surén and colleagues used a prospective nationwide cohort study for children born in Norway between 1999 and 2009. Of these, 85,176 agreed to participate and met criteria, and 114 were identified as having been diagnosed with autistic disorder. FA status was determined via self-report of supplement use completed by the mothers at 18 weeks gestation. At the time of the study, food items were not additionally supplemented in Norway. Additional questioning about diet and supplement use was conducted at week 22. The interval of interest for FA intake was a three-month period from 4 weeks BEFORE conception to 8 weeks after conception. Logistic regression was used, and the effects of parental education, parity, whether the pregnancy was planned, smoking, and year of birth were all considered as potential confounders. Of these, birth year, education of mother, and parity were found to predict ASD, and were controlled. FA near the time of conception was associated with a significant reduction of risk of developing autism (OR = 0.51 unadjusted for confounds, remaining significant with an OR = 0.61 after adjusting for confounds.

### 5.2. Schmidt et al., 2012 [9]

A large, well-vetted database from Northern California and case-control methodology was used. Children were born after 1999, and therefore all participants would be born after food supplementation began. Parents were interviewed and asked about their use of supplements as well as other questions. The time of interest for this study was defined as three months before conception to one month after conception. Even after adjusting for maternal education and birth year, supplementation with prenatal vitamins (which contained FA) was associated with a decreased risk of autism.

### 5.3. Raghavan et al., 2016 [15], Raghavan et al., 2017 [49]

Two recently published studies from this group have reported on ASD risk as it correlates with FA and vitamin B12 levels at the time of delivery. The 2016 study [15] was a prospective study and used the Boston birth cohort, which included 1391 births and 107 cases of autism. This is the only study that uses measured levels of folates rather than self-report of supplementation. Blood samples from the mother were obtained within 72 h of delivery, and later, the total level of plasma folate was obtained. Specifically, the predictor variable was blood measures of folate measured at the time of birth, and the outcome of interest was a subsequent diagnosis of autism in the offspring. High blood folate status was associated with a doubling in risk for ASD. Ten percent of the sample had what was considered an excess amount of folate in the blood at the time of delivery.

The 2017 study [49] included data from the same cohort that included 1257 mother-child pairs (showing 86 cases of ASD), but also included survey-interview data from the mothers on their multivitamin supplementation. The levels of supplementation were divided into low (two or fewer times per week), moderate (three to five times per week) and high (more than five times per week). The authors reported that both low and high supplementation were associated with increased risk for ASD, and that very high maternal blood levels of folate at the time of birth, and very high levels of maternal blood vitamin B12 at birth both increased the risk of ASD 2.5 times [49]. In these reports, it is apparent that plasma folate or B12 levels were the variable of interest, rather than unmetabolized FA, yet the risk was elevated, and multivitamin supplementation that is too low or too high correlated with the elevated risk.

### 5.4. DeSoto & Hitlan, 2012 [14]

This research used the well-vetted Centers for Disease Control data set [48], and used a case-control design. Included were 256 children classified as having an ASD and 752 controls born between 1994 and 1999. A stratification procedure was employed such that participants were stratified by gender, HMO, and year of birth by design; thus, a conditional logistic regression was used for data analysis. A conditional logistic regression was employed to check for the effect on diagnosis while adjusting for several covariates (maternal age, birth weight, poverty ratio, birth order, maternal prenatal healthcare-seeking behavior and child medical conditions). ASD was positively related to self-report of folic acid supplementation. Considering only autistic disorder per se, positive relations remained between autism diagnosis and folic acid supplementation. Overall, for ASD, after adjusting for other covariates (including health-seeking behavior) a significant effect of folic acid supplementation on ASD diagnosis remained (χ^2^ = 5.37, *p* = 0.020, OR: 2.34, 95% CI: 1.14–4.82). The Odds Ratio of 2.34 indicates that mothers who self-report using folic acid supplementation were more than twice as likely to have a child with ASD.

### 5.5. The Autism Risk Studies

A lack of vital nutrients in early pregnancy is clearly a problem. A general protective effect for prenatal vitamins and adequate amounts of folates especially during the preconception period and early pregnancy are clearly established, and seem beyond dispute [6,7]. The two studies that report a protective effect of high pharmacologic concentrations of FA both used the preconception and very early pregnancy as the window of interest, and employed the use of prenatal vitamins as the predictor. To our knowledge, no studies have found a protective effect for FA above recommended dosages for middle or late pregnancy, and two have found evidence of a harmful effect. This is consistent with animal research, which shows negative effects of gestational FA supplementation, as well as in vitro research showing that unmemetabolized FA impairs neurite and growth cone development and synaptogenesis (both reviewed above). It seems possible that prenatal vitamin benefits in early development (such as protection from neural tube defects) prevent harmful effects associated with specific lack of key nutrients. However, this does not mean that sustained high intake of FA and unmetabolized FA circulating throughout pregnancy well into the third trimester is without risk. During the final trimester, the brain triples in size, and the cerebral cortex undergoes most of its development. In this view, the benefits of early supplements remain; however, caution is warranted regarding the sum amounts of folic acid accumulating from cereals and grains combined with prenatal supplements. We advise caution in order to avoid the outcome of 10% of mothers who may accumulate FA at excess levels [14].

## 6. Discussion

It has been widely believed that multivitamin supplements that contain FA clearly prevent neural tube defects. As a result, in 1993, it became the official recommendation in the United States that all women of childbearing age take FA as a supplement. In 1998, the US and Canada took the additional step of requiring that all grain products sold have significant amounts of supplemental folic acid added. As a result, over the past generation, there has been a major shift upwards in human intake of FA. FA is not identical to folate from naturally occurring food sources. Nonetheless, this has sharply reduced the number of neural tube-related birth defects. The benefits associated with supplying sufficient vitamins during the weeks around conception are clear.

Conversely, continual high levels of FA supplementation throughout a pregnancy may not be needed, and are not without risk. The role of folates in synthesizing nucleotides and in methylation reactions as a methyl donor is fundamental to virtually all aspects of development and health. However, unexpectedly high levels of FA may have inadvertent implications for proper methylation of DNA during times of rapid cell division, such as in prenatal development. The results of Barua and colleagues in 2014–2016 (reviewed above) clearly document the potential for high FA supplementation to alter genomic functioning and affect behavior. The idea that adding folic acid to the food supply might have unintended consequences has been articulated as early as 2005, and was specifically speculated to be relevant for the increase in autism in 2011 [18]. Generally speaking, the results of recent research on supplementation with FA suggest supplementation may have unintended negative consequences, and that selective excess intake of one vitamin type may have the potential to negatively alter metabolic activities. The tolerable safe upper limit of FA has been suggested to be 1000 mcg (μg) per day. Prenatal vitamins typically contain 1000 mcg. To put this in perspective, a bowl of breakfast cereal alone may have 400 mcg of supplemented folic acid. Data from the National Health and Nutrition Examination Survey (NHANES 1999–2000) documents how rapidly the increase occurred. For example, since an earlier NHANES study (conducted between 1988 and 1994), the average blood levels increased from 12.5 nmol/L to 32.2 nmol/L. Persons with levels in the high range increased from 7 to 43% in this short time. Finally, recent surveys suggest that more than 10% of pregnant women are taking folic acid dietary supplements in excess of 1000 mcg per day while pregnant [59], not including the supplementation they are ingesting if they eat cereal, bread, or pasta.

Adding to these concerns is the recognition that for treatment and prevention of ASD, FA is not the correct folate to use. Promising new understanding of functional metabolic blockages in folate pathways has emerged [50,51], and has led to successful treatment with folinic acid [53]. Furthermore, the prevalent existence of autoantibodies (and accompanying inflammation) that interfere with folate transfer into the brain, has become apparent from research [55,56]. Thus, new, more appropriate treatments are becoming possible.

Further research on these issues will be forthcoming and important. However, although optimal levels are important for development, there is no known benefit to exceeding the RDA for folates, and yet this is occurring. It is clear that some women are taking more than 1000 mcg a day while they are pregnant. There is experimental animal and in vitro research documenting negative effects of excess or unmetabolized FA on genetic programming and neuronal development. At least three studies have suggested that high levels of supplementation when taken throughout pregnancy may be associated with negative neurodevelopmental outcomes in offspring [14,15,16]. As a whole, caution regarding over supplementing is warranted.

## Figures and Tables

**Table 1 brainsci-07-00149-t001:** Five important empirical findings related to Folic Acid supplementation.

FA Supplementation Actual Intakes are Higher than Were Projected When Supplementation Was Instituted.	Quinlivan & Gregory, 2003 [29]
Unmetabolized FA occurs in cord blood, and in a dose dependent manner in serum when FA supplementation is more than about 200 μg/d.	Sweeney at al., 2006 [25]; Kalmbach et al., 2008 [24]; Troen et al., 2006 [27]; Obeid et al., 2010 [28]
Evidence from rodent studies showed that FA exposure during gestation caused changes in gene expression and anxiety, and hyperactivity in offspring. Exposure to higher-dose FA diet during gestation caused dysregulation of expression of many genes, including neuro-developmental genes, and epigenomic changes.	Barua et al., 2014 [38]; 2015 [39], 2016 [40]
Some women exceed recommended levels of FA during pregnancy.	Hoyo et al., 2011 [48]; Raghavan et al., 2016 [15], 2017 [49]
Folic acid can inhibit neurite extension, growth cone activity and synaptogenesis.	Wiens et al., 2016 [41]

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
