# Peer review of "Is High Folic Acid Intake a Risk Factor for Autism?—A Review"

_brainsci, 2017, doi:10.3390/brainsci7110149_

Round 1

Reviewer 1 Report

This is a well-written review and discussion of the literature on possible risks with high folic acid consumption during pregnancy, notably the association with autistic syndrome in the offspring. I find the paper basically good and interesting. My only concern is the unlimited assurance that folic acid when taken periconceptionally decreases the risk of neural tube defects, many times repeated in the text. The references given mainly refer to multivitamin use and not to folic acid use. A recent critical discussion of the literature linking folic acid use with a decline in neural tube defect rate throws some doubt on the widely accepted idea that this has been a great success (Kallen B: Folic Acid and Neural Tube Defects. Effective Primary Prevention or Wishful Thinking? Avid Science Monograph Series, 2017). My suggestion is that the authors modify their statements, perhaps instead of stating that the effect is certain say that it is generally believed or accepted that folic acid prevents neural tube defects.

Author Response

The reviewer raises a question about the widely held assertion that folic acid itself decreases the risk for or can prevent neural tube defects.  We had made several statements in the manuscript that do ascribe to this view.  We believe the reviewer is correct in questioning the complete validity of this, even though it widely accepted.  The reviewer provided a reference to a monograph by Kãllén published just this year that thoroughly examines this issue.  We have now included this reference (page 1, line 43) and we have modified our statements on page 1, lines 13, 39, 305, to acknowledge that the assertion has been and continues to be questioned despite widespread acceptance.  We thank the reviewer for these comments.

Reviewer 2 Report

In the manuscript submitted to Brain Science (229634v1) authors show a review on the evidence that high folic acid is a risk factor autism. This reviewer suggest the publication in Brain Science after minor revision.

Theme is interesting and review well performed. Information could be improved if authors provide a Table more complete, including data of most of the cites than appears in the manuscript.

Author Response

This reviewer has only the one suggestion for improvement:  that we make Table 1 more complete so as to include more of the citations that we have in the text.  Therefore we have added three more citations that are relevant.

Reviewer 3 Report

The major problem is that the review misses the problems with folate metabolism that have been found recently and the implications. Specifically, the issue is that the wrong type of folate is being used to supplement. That is, many people have blocks in folate metabolism that make folinic acid, the synthetic form, not metabolized. This results in increased levels of folinic acid in the blood which is a biomarkers of poor folate metabolism. That is, oxidized folate (folinic acid) is not being converted into a reduced folate usable by the body. In addition, the review misses the entire literature on the folate autoantibody which blocks folic acid into the brain and across the placenta. Thus, in the context of the folate autoantibody, folinic acid, which is commonly used to supplement folate, is not the preferred form of folate and reduced folate, such as folinic acid (See recent animal models by Quadros), can rescue those with blocks in folate metabolism.. 

Author Response

The reviewer points out that our review paper has missed two important areas of research, namely the recent studies that have shown linkages of autism with abnormalities in metabolic physiology, and a body of literature dealing with the phenomenon of autoantibodies to the folate receptor, both blocking antibodies and binding antibodies that evoke immune reaction in patients.  We agree that these areas are important, are within the scope of our review, and that the discoveries have implications for the public health issue of supplementation and fortification with folic acid.  Therefore we have written a new section entitled “Metabolic Abnormalities and Autoantibodies in Autism”.  We are including this in lines 176-208, between the section on “Neural Development” that ends with Table 1, and the “Autism Risk Studies” section that follows it.  We thank the reviewer for bringing this to our attention.  The addition improves the paper.

Reviewer 4 Report

The manuscript by Wiens and DeSoto is a review of previous publications on the risk / benefits of high dose folic acid supplementation in pregnancy and in the general population. Perhaps a more appropriate title should be “Is high dose folic acid intake a risk factor for autism?” This will allow the authors to provide a balanced review of data and arguments from both sides of the discussion. The review provides data from dietary supplementation as well as from prenatal prescription intake, thus mixing both normal low dose intake and high dose intake. These are best analyzed separately in the review.

Studies of valproic acid have no place in this review since that brings in blocking of folate metabolism and other toxic effects into play. The effects of VA on folate metabolism have to be considered.

Similarly, the in vitro chick embryo studies are difficult to reconcile with the findings because the effect of other forms of folates and mono/polyglutamates are missing from these studies. Throughout the manuscript, the terms folic acid and folate have been used. It is not clear to me where actual measurements folic acid and other forms of folate were made and where it is loosely used as a generic term. The authors should review published data and report quantitative values in each study for comparsion of the actual levels reported. For example, table 1 should include values for folic acid and total folate and time frame for the analysis done after the last dose. While some data is provided in the autism risk section, it is incomplete. The section on unmetabolised folic acid should include original quantitative data for the reader.

The Raghavan study reported indicates excess folate. Do the authors mean excess folic acid in the blood? A careful review and reporting of published data is warranted. I get the impression that the risk is associated with folic acid intake and not necessarily with unmetabolized folic acid. Line 298 indicates 1000mg of folic acid intake by pregnant women. I am not aware of such high dose.

Author Response

Reviewer 4 begins by noting that this is a review of research on folates and risk/benefit specific to autism and suggests an alternative title. We agree with reviewer’s reasoning and that a new title may be more focused to the content. The title has been changed to, “Is high folic acid intake a risk factor for autism?—a review”

Reviewer 4 next states, “The review provides data from dietary supplementation as well as from prenatal prescription intake, thus mixing both normal low dose intake and high dose intake,” and suggests these might best be separated in the review.  The concern the manuscript addresses is that the sum total of folic acid may exceed optimal levels and could cause harm if and when this happens.  Such high amounts could occur because, today, the vast majority of women who are pregnant take prenatal vitamins with folic acid, and consume grains that are also fortified with folic acid. It is the sum that is of interest.  We are not sure how separating existing research would work other than what we have reviewed.  We agree that more research is needed with more careful measures, and hope this review might spur such research on.  Prenatal vitamin are recommended to take, and can be assumed to have at least 400 mcg of folic acid.  Suren 2013, Schmidt 2008, and Desoto 2012 based their tests of autism risk on vitamin taking compared with those who did not, with the assumption that those who took the supplements are getting higher levels of FA. Dietary supplementation happens for all women (assuming they injest any flour or bread, or cereal or grains, rice, oats), and most then take prenatal vitamin supplements also.  Thus, it might seem that the only way to “separate these” would be to compare those who do with those who do not take the vitamin supplements while pregnant.  In sum, we are not sure we fully understand this recommendation.

The reviewer states that studies of valproic acid have no place in this review because of possible effects it may have on folate metabolism.  We are willing to accept this view in the context of this review’s subject.  Therefore the single paragraph that dealt with it on page 3 has been removed.

Another issue this reviewer raises is the relevance of the in vitro studies using chick embryo neurons since only FA and not other forms of folates were used.  However, in the Wiens et al (2016) study, folinic acid was indeed included in the study of effects on neurite length, and it was found not to have the inhibitory effect that FA showed.  It was not studied in a concentration profile as this paper was focused on FA effects, and this is probably the reason it was not noticed.  We believe that the in vitro study described in the review is important to include because it is the only such study of the direct effects of FA on neurons themselves.  We agree that further research is needed. We think the existing studies, as a whole, strongly suggest the need for more research and hope that this review will lead to additional studies. We plan to study the effect of natural folate compared with folic acid in future research, and agree with the reviewer this is an important step. The point is well taken.  Here, the existing research is showing that unmetabolized folic acid—if it circulates and makes contact with developing neurons-- has the potential for alter normal neural development. We think this in itself is also important (especially since the report of Sweeney et al. 2006 shows that unmetabolized FA is now found in cord blood).

Reviewer 4 states: “ Throughout the manuscript, the terms folic acid and folate have been used. It is not clear to me where actual measurements folic acid and other forms of folate were made and where it is loosely used as a generic term.”  This has been clarified throughout the MS.  Folatesrefers to those forms naturally present in green leafy vegetables, nuts and dairy products; grain-based foods are supplemented with FA. They are not identical and the reviewers point is well taken.  We added the clarification “of FA” to the sentence in what is now line 305-307 in the Discussion.  The subject of this sentence now reads:  “However, unexpectedly high levels of FA may have inadvertent…

Reviewer 4 makes an important point about using quantitative data for purposes of comparisons:  “The authors should review published data and report quantitative values in each study for comparsion of the actual levels reported.  For example, Table 1 should include values for folic acid and total folate and time frame for the analysis done after the last dose. While some data is provided in the autism risk section, it is incomplete. The section on unmetabolised folic acid should include original quantitative data for the reader.”

Table 1 contains a mixture of research findings. For the most part, it is not possible to include precise values for folic acid and total folate and ‘time frame after the last dose’ for the various research findings reported in Table 1. This detail could (at least partly) be done for the animal research reported IF deemed truly crucial. But we tend to think this information could be found within the referenced article if a given reader needed this information.

However we do agree on the use of quantitative data where possible.  Therefore, we have added additional quantitative data to the review’s section “Unmetabolized FA” as Reviewer 4 suggests.  We believe the paper is improved through this addition and we thank the reviewer for this suggestion. The section now reads (beginning at line 70) as follows:

Because there are individual differences in DHFR activity in humans (e.g. C667T variants), and because humans, as a species, have low activity of this enzyme, the competition for binding enzyme is potentially relevant [22], especially so in some persons.

Dietary supplementation coupled with pervasive fortification of grain-derived foods with synthetic FA beginning in the late 1990’s may have created a demographic with high serum levels of unmetabolized FA and high erythrocyte FA concentration [summarized in 19, 23].  This is not theoretical, but has been demonstrated in a dose dependent manner when levels of intake are over 200 micrograms per day [24]. FA is detectable in fasting serum of a majority of subjects tested and the proportions with detectable levels have risen since fortification programs were initiated [25, 26], even becoming detectable in umbilical cord blood [27].  When food supplementation began, models suggested that supplement levels would be approximately 100 micrograms per day, but the actual increasehas exceeded 215 mcg, double that estimate [28].  Consumption of more than 1 mg of folic acid appears to reliably result in unmetabolized FA, even if the doses are spaced apart. Sweeney andcolleagues[25]investigated the effects of consumption usingvaryingdosage schedules. The cumulative amount is what mattered. They administered FA to participants in five equal doses (of 200 mcg) across a day. All participants showed unmetabolized FA circulating after the second, third, fourth and fifth doses, with the highest levels(up to 5 mcg/l)after the final dose [25]. This raises concerns about detrimental effects of high serum synthetic FA [30].  These include effects on the enzyme dihydrofolate reductase [22], regulation of folate uptake in renal and intestinal epithelia [31], reduced cytotoxicity of natural killer T cells in postmenopausal women [27], disregulation of gene expression in lymphoblastoid cells [32], and cytotoxicity to neural tissues and mental health [reviewed in 33, 34].  In addition, evidence shows that high FA intake is associated with an increase in incidence of twin births, body fat mass and insulin resistance in offspring, increased risk of colorectal cancer, and other adverse outcomes [reviewed in 20].

Finally, Reviewer 4 commented:  “The Raghavan study reported indicates excess folate.  Do the authors mean excess folic acid in the blood?  A careful review and reporting of published data is warranted.  I get the impression that the risk is associated with folic acid intake and not necessarily with unmetabolized folic acid.  Line 298 indicates 1000 mg of folic acid intake by pregnant women.  I am not aware of such high dose.”

The methods used in the Raghaven et al study measured total folates in the blood at delivery and did not specifically differentiate subtypes (e.g., pteroylmonoglutamic acid vs formyl tetrahydropteroylglutamate).  Yes, the authors measured blood levels.  We seek to clarify and we have changed the sentence in lines 251-252 and 247-249 (changes here shown in red).  This paragraph now reads as follows:

This recently published research was a prospective study and used the Boston birth cohort, which included 1391 births and 107 cases of autism. This is the only study that uses measured levels of folates rather than self-report of supplementation.  Blood samples from the mother were obtained within 72 hours of delivery and later the total level of plasma folate was obtained.  Specifically, the predictor variable was blood measures of folate measured at the time of birth, and the outcome of interest was alaterdiagnosis of autism in the offspring. High folate was associated with a doubling in risk for ASD.  Ten percent of the sample had what was considered an excess amount of folate in the blood at the time of delivery.

Reviewer 4 indicates that she/he is not aware of reported intake doses as high or higher than 1000 mg per day for pregnant women.  We believe this is meant to be micrograms (µg or mcg) per day.  We supported our report of 1000 micrograms (mcg) with citation 57.  It seems that we neglected to add the reference for this.  It was reported in the study by Hoyo et al. 2011.  We have now added this reference as number 58.

Round 2

Reviewer 3 Report

The authors have done a great job at addressing concerns of the reviewers. However, I think they fall short of making the point that alternative types of folate, such as reduced forms of folates, may be a solution to prevent the unmetabolized folic acid. In this sense the last paragraph of the paper just states that over supplementation is bad, but this because FA is the wrong type of folate. In addition, knowing the functional blocks in the folate pathway may be helpful for determining the type and amount of folate to be supplemented. In this sense the authors should make it clear the alternative types of folate may be very useful in the context they discuss. I would suggest stating at the end of the abstract and the end of the paper that alternative types of folate, such as reduced folates, may be the solution to the problem the authors discuss

Author Response

Reviewer 2

The authors have done a great job at addressing concerns of the reviewers. However, I think they fall short of making the point that alternative types of folate, such as reduced forms of folates, may be a solution to prevent the unmetabolized folic acid. In this sense the last paragraph of the paper just states that over supplementation is bad, but this because FA is the wrong type of folate. In addition, knowing the functional blocks in the folate pathway may be helpful for determining the type and amount of folate to be supplemented. In this sense the authors should make it clear the alternative types of folate may be very useful in the context they discuss. I would suggest stating at the end of the abstract and the end of the paper that alternative types of folate, such as reduced folates, may be the solution to the problem the authors discuss

Response:  We agree with Reviewer 2 and acknowledge the oversight.  To remedy this, we have altered the last portion of the abstract to:

“In this review we outline the reasons excess FA supplementation is a concern and review the history and effects of supplementation.  We then examine the effects of FA on neuronal development from tissue culture experiments, review recent advances in understanding of metabolic functional blocks in causing ASD and treatment for these with alternative forms such as folinic acid, and finally summarize the conflicting epidemiological findings regarding ASD.  Based on the evidence evaluated, we conclude that caution regarding over supplementing is warranted.”

In addition, we have added a new paragraph that deals with the issue to the discussion near the end.  It comes just before the very last paragraph.  It reads:

Adding to these concerns is the recognition that for treatment and prevention of ASD, recent research suggests that FA is not the correct folate to use.  Promising new understanding of functional metabolic blockages in folate pathways has emerged [48, 49], and it has led to successful treatment with folinic acid (51).  Furthermore, the prevalent existence of autoantibodies (and accompanying inflammation) that interfere with folate transfer into the brain has become apparent from research [53, 54].  Thus new, more appropriate treatments are becoming possible. 

Reviewer 4 Report

The manuscript is substantially improved. However, some additional revision is needed.

1. The abstract needs a concluding statement “ Based on the evidence evaluated we conclude that------???

2. Line 65 statement does not have a reference.  My review of literature indicates that at physiologic or at  somewhat higher does, almost all of the folic acid is converted to  methyfolate in the gut.

3. The authors should play down the results of the in vitro studies   and the rat  studies because the concentrations used should be  considered extremely high and should not be extrapolated to levels seen  in people taking vitamin supplements.

4. Line 184 should read” folinic acid has been used to correct the folate deficiency in CFD”

5. Line 201 should be corrected to “ exposure to rat FRalpha antibodies----“

6. In quoting the Raghavan study, the authors should include their recent publication shown below  and clearly indicate that the study does not address  unmetabolized folic acid but rather folate status at the time of birth.  There could be another reason for the outcome in mothers with elevated  folate. These mothers were taking higher doses because  they had a higher risk of having an autistic child and that even higher  folate intake did not prevent this.

Paediatr Perinat Epidemiol.  2017 Oct 6. doi: 10.1111/ppe.12414. [Epub ahead of print]

7. Line 278 statement needs a reference

8. Line 279 should indicate high pharmacologic concentrations of folic acid.

Author Response

Reviewer 2

The authors have done a great job at addressing concerns of the reviewers. However, I think they fall short of making the point that alternative types of folate, such as reduced forms of folates, may be a solution to prevent the unmetabolized folic acid. In this sense the last paragraph of the paper just states that over supplementation is bad, but this because FA is the wrong type of folate. In addition, knowing the functional blocks in the folate pathway may be helpful for determining the type and amount of folate to be supplemented. In this sense the authors should make it clear the alternative types of folate may be very useful in the context they discuss. I would suggest stating at the end of the abstract and the end of the paper that alternative types of folate, such as reduced folates, may be the solution to the problem the authors discuss

 Reviewer 4 (Our responses are in blue type following each comment.)

The manuscript is substantially improved. However, some additional revision is needed.

1. The abstract needs a concluding statement “ Based on the evidence evaluated we conclude that------???

This suggestion is very good; we agree that a concluding statement improves the abstract a great deal.  The last sentence now reads:  “.  “Based on the evidence evaluated, we conclude that caution regarding over supplementing is warranted.”

2. Line 65 statement does not have a reference.  My review of literature indicates that at physiologic or at  somewhat higher does, almost all of the folic acid is converted to  methyfolate in the gut.

Our reference for the statement is Powers 2007 which is number 23 in our references list, and we have now inserted it following the statement.  It is the statement concerning the initial reduction and methylation of folic acid being mainly in the liver rather than in the intestine.  The commentary by Powers 2007 cites Rogers et al (1997) and Wright et al (2005) papers, both published in the Journal of Nutrition, and these provide detailed information on the initial metabolic processing .  If the reviewer requests it, we can add these as well but we think the Powers 2007 reference is sufficient.

3. The authors should play down the results of the in vitrostudies   and the rat  studies because the concentrations used should be  considered extremely high and should not be extrapolated to levels seen  in people taking vitamin supplements.

In response to this comment, we have added cautionary sentences to the Neural Development section.  In the case of the rodent studies, the new sentence reads:

However, these studies using rodents saw the effects with maternal diets in a range of FA approximately ten-fold higher than is recommended for normal pregnant women (a level chosen because women with a history of NTD affected pregnancy have been prescribed supplementation at ten-fold higher).  In the case of the in vitro study, the new sentence reads:  Although this study used FA concentrations that are (as with the rodent studies) quite high, and should not be extrapolated to the levels observed in humans, the findings support the idea of a theoretical direct effect.

4. Line 184 should read” folinic acid has been used to correct the folate deficiency in CFD”

We acknowledge this more accurate phrasing and have made the suggested change.

5. Line 201 should be corrected to “ exposure to rat FRalpha antibodies----“  We have made this suggested correction.

6. In quoting the Raghavan study, the authors should include their recent publication shown below  and clearly indicate that the study does not address  unmetabolized folic acid but rather folate status at the time of birth.  There could be another reason for the outcome in mothers with elevated  folate. These mothers were taking higher doses because  they had a higher risk of having an autistic child and that even higher  folate intake did not prevent this.  We have added a new paragraph under the subtitle “Raghavan et al.” and in this we have summarized the findings of the Raghavan et al. 2017 study (which is just published this month online).  We have added the emphasis suggested by the reviewer.  The new inserted paragraph reads as follows: 

The 2017 study [57] included the same kind of data from a sample of 1257 mother-child pairs (showing 86 cases of ASD), but also included survey-interview data from the mothers on their multivitamin supplementation.  The levels of supplementation were divided into low (two or fewer times per week), moderate (three to five times per week) and high (more than five times per week).  The authors reported that both low and high supplementation were associated with increased risk for ASD, and that very high maternal blood levels of folate at the time of birth, and very high levels of maternal blood vitamin B12 at birth both increased the risk of ASD 2.5 times [57].  In these reports, it is apparent that plasma folate or B12 levels were the variable of interest rather than unmetabolized FA, yet the risk was elevated, and multivitamin supplementation that is too low or too high correlated with the elevated risk.

Paediatr Perinat Epidemiol.  2017 Oct 6. doi: 10.1111/ppe.12414. [Epub ahead of print]

7. Line 278 statement needs a reference.  We have supplied two citations for this statement.  These were already present in the reference list.

8. Line 279 should indicate high pharmacologic concentrations of folic acid.  We have added this insertion.